# Space Reproduction in Urban China: Toward a Theoretical Framework of Urban Regeneration

**Yafei Liu**

Shanghai Tongji Urban Planning and Design Institute, 1111 Zhongshan North Second Road, Shanghai 200437, China; liuyafei@tjupdi.com

**Abstract:** As China's urbanization enters the middle and late stages, urban regeneration has risen to the strategic level of building a new national development pattern and promoting high-quality urban development. Due to the wide range of disciplines and content involved in urban regeneration, there remains a lack of systematic and comprehensive theories and frameworks to lay a theoretical foundation for academic research and provide guidance for renewal practice. Therefore, this paper aims to construct a systematic and comprehensive theoretical framework of urban regeneration from the perspective of space reproduction, by employing the theory of the production of space as the basis of interdisciplinary research, and integrating related theories and research. The current paper proposes a theoretical framework that includes four core elements, namely the government, the market, society and space reproduction. Subsequently, the paper summarizes the subject, power, capital and interest of the three types of stakeholders (government, market, and society), their different emphases in the reproduction mechanism, and their various cooperative modes in practices. Meanwhile, under the concept of sustainable development and "people-oriented" approach, this paper emphasizes the importance of social factors and the suitability of the multi-stakeholder co-governance model of government, market and society.

**Keywords:** urban regeneration; space reproduction; theoretical framework; capital proliferation; multi-stakeholder co-governance

## 1. Introduction

China's reform and opening up policy, launched in 1978, introduced private business and market incentives to what was a state-led communist system, bringing about an unprecedented economic boom. The speedy growth of China's economy has also driven the rapid construction and development of cities [1]. In just 40 years, the urbanization rate has risen from 17.9% in 1978 to 51.2% in 2011 and 64.7% in 2021. China has entered the middle and late stages of urbanization, and stepped into a transition period. With the transformation of China's economy from high-speed growth to high-quality development, the urban development mode has also shifted from extensive and extensional to intensive and connotative development, and gradually from large-scale incremental construction to the improvement of existing urban quality and urban regeneration. National ministries and local governments, such as Shanghai, Shenzhen and Guangzhou, have successively issued a number of laws and regulations to guide urban regeneration in recent years [2–4]. The Fifth Plenary Session of the 19th Central Committee of the Communist Party of China in 2020, and the National Two Sessions in 2021 and 2022, have clearly put forward the "implementation of urban regeneration actions", which has raised urban regeneration to the strategic level of building a new national development pattern and promoting high-quality urban development.

Urban regeneration is a comprehensive and integrated vision and action which seeks to resolve urban problems and continuously improve the economic, physical, social and environmental condition of an area that has been subject to change or offers opportunities

for improvement [5]. Recent years have witnessed many studies and discussions on the concept and connotations, dynamic mechanism, stage characteristics, stakeholders, targets and principles, planning implementation, operation, and performance of urban regeneration in Chinese and international academic circles [6–12]. The essence of urban regeneration can be understood as the re-creation of space value and the redistribution of new benefits through the readjustment of space resources. The stakeholders who promote urban regeneration usually include the central and local governments, developers, and the occupiers and users of the renewal space. The spatial resources and specific objects involved in urban regeneration commonly comprise old residential areas, old industrial areas, old commercial areas, shanty areas, urban villages, and historical districts. Regarding the stage characteristics of urban regeneration in the past 100 years, Chinese and Western cities have generally experienced a transformation process from focusing on the reconstruction and renewal of the physical environment to also taking into account economic, social and environmental goals and urban governance [13,14]. Through urban regeneration, the re-creation of spatial value is realized. For instance, the quality of the living environment has been improved, urban brand and competitiveness have been strengthened [15], industrial upgrading and economic revitalization have been promoted, and the value of land and real estate has increased. With regard to the redistribution of new benefits, the international regeneration practice in the past 100 years has also shown a certain regularity, from the model of the government leading reconstruction and renewal and citizens enjoying the improvement of the living environment, to the government and enterprises forming an urban growth alliance under the neoliberal or market economy period and jointly promoting urban regeneration and sharing value-added benefits [16,17], and then to a mode of advocating humanism, sustainability, "government-enterprise-society" multi-cooperation and benefit sharing [18–20].

It can be seen that the content and scope of urban regeneration has been quite extensive, and involves many disciplines, such as institutional economics, sociology, public management, law, geography, urban planning and architecture. However, this also reflects some bottlenecks faced by the current research and practice. First of all, regarding the academic research, although there are many related theories involved in urban regeneration, such as sustainable development theory, urban marketing theory, social justice theory, space production theory, differential land rent theory, etc., a recognized theory of urban regeneration that can comprehensively explain its rich connotations and complicated mechanism has not yet been developed [5]. Without a systematic and integrated theoretical framework as a basis, academic research might encounter difficulties in comprehensively investigating the complex regeneration mechanism, and might draw very partial, one-sided, and even wrong conclusions. Secondly, for the renewal practice, the lack of a systematic and comprehensive theoretical framework could jeopardize the soundness of the science and effectiveness of the top-level renewal policies [6], which might in turn introduce errors in goal-setting and benefits distribution or weaken the performance of implementation and operation in practice.

Therefore, this paper aims to integrate the relevant theories and research studies on urban regeneration, and construct a systematic and comprehensive theoretical framework. Since urban regeneration refers to the improvement of physical space and its attached economic and social environment, space plays a crucial role not only as the direct object of urban regeneration, but also as the product and carrier of economic and social activities. Hence, space can be employed as a core element in the theoretical framework of urban regeneration and as a link to integrate other elements. Since the theory of the production of space, originating from political economy, provides a unified perspective for complex urban social science research (including urban sociology, urban geography, urban planning, etc.), it has gradually received more attention in urban regeneration research [13,21–23]. Therefore, the current paper takes the theory of the production of space as the basis of this interdisciplinary research, and constructs a theoretical framework of urban regeneration from the perspective of space reproduction by studying related theoretical frameworks

and renewal practices. Finally, the theoretical and practical implications of this framework are discussed.

## 2. Methodological Approach

In accordance with the aim of this paper, a literature review is employed as the major methodology to overview relevant research fields, to track the research development over time, and to identify key components for building a new theoretical model. To fulfill the various needs of this paper, a mixed literature review approach is utilized which consists of the following two broad types: the semi-systematic (or narrative) review approach, and the integrative (or critical) review approach [24]. Firstly, considering the interdisciplinary characteristics of urban regeneration, the semi-systematic review approach is chosen which is especially designed for the topics that have been studied by researchers from diverse disciplines [24,25]. The semi-systematic review approach is used to synthesize the state of knowledge of urban regeneration, to detect its themes and theoretical perspectives, to identify key components, and to provide a historical overview or timeline. Secondly, rather than simply providing an overview, this paper aims to generate a new comprehensive theoretical framework. Therefore, the integrative review method is used to critically analyze and examine the key components of urban regeneration and their relationships, and to eventually promote the advancement of this theoretical framework [24,26].

In the first section of this paper, the state of knowledge of urban regeneration is reviewed, and the research gap is identified. The third section reviews the theory and relevant research of space production and reproduction, including the key theoretical components, (re)production mechanism, (re)produced built environment, etc. Then, a historical overview of urban regeneration in Western and Chinese cities from a space reproduction perspective is provided, including the background, goals, key stakeholders, cooperating mechanism, reproduced urban space, benefit distribution, etc. Thereafter, the fourth section reviews the literature particularly focused on a theoretical framework with a space reproduction perspective. The review of the literature identifies the key components, stakeholders, powers and capital, interests, operating mechanisms, game relationship, etc., within the theoretical framework. Meanwhile, using an integrative or critical review style, some drawbacks of existing theoretical framework are suggested. By synthesizing and integrating all the relevant knowledge and information, the fifth section suggests a systematic and comprehensive theoretical framework.

The literature review was conducted by searching for peer-reviewed articles in search engines of Scopus and CNKI (China National Knowledge Infrastructure). To synthesize the literature, a broad range of keywords from diverse disciplines are used to identify relevant papers, including the following: urban regeneration, urban renewal, urban redevelopment, production of space, space reproduction, theoretical framework, conceptual framework, etc. These search terms are closely associated with the purpose, scope, gap and research question the review aims to address, which act as the inclusion criteria. Moreover, regarding other inclusion criteria, special attention is paid to the theoretical and conceptual articles, existing review articles, articles published in recent years, journal articles, and articles focused on the Chinese context.

## 3. Understanding Urban Regeneration from the Space Reproduction Perspective

In general, the connotation of space production in this paper is similar to urbanization, while space reproduction is closely associated with urban regeneration. More importantly, the theory of space (re)production provides a systematic and appropriate perspective to help us understand the complex mechanism of urbanization and regeneration. This part firstly discusses urban space and its changes driven by the underlying capital and power from the space reproduction perspective. Thereafter, such a perspective is employed to re-analyze and re-understand the urban regeneration process in Western and Chinese cities over the last century.

### 3.1. Space Production and Space Reproduction

Space production can be understood as a perspective or way of thinking about and analyzing urban space. Different from only emphasizing the materiality of space and treating it as a container and site, the theory of the production of space explores the social and dynamic nature of space, arguing that space is the product of social production, and space production is a dynamic process in which political and economic elements, such as capital and power, shape and transform urban space [27–30]. The French Marxist thinker Henri Lefebvre first proposed the theory of the production of space in the 1970s, and constructed a theoretical framework including spatial practice, representations of space, and representational spaces [27]. Among these concepts, spatial practice, focusing on the perceived physical space, refers to the social production and reproduction practice in daily life implemented by the socio-spatial unity of social groups and their spatial carriers. Examples, in this regard, include the formation of suburban middle-class neighborhoods, and the gentrification of old central cities. The representations of space, focusing on the conceived abstract space, are the abstract, conceptualized spatial order and system dominated by the knowledge and ideology of politicians, scientists, planners, and technologists, e.g., the urban development blueprints and spatial system schemes proposed by urban planners for the government. The representational spaces, focusing on the intentional space in life, refer to the symbolic, imaginary, emotional, and historical and cultural space superimposed on the physical space by the space users, with an example being the urban image in citizens' minds about city centers, landmarks, childhood homes, historic districts, etc. In the process of space production, the participating social groups and stakeholders use the capital and resources at their disposal to carry out spatial practice based on their conception, planning and control of the representations of space, and eventually shape the representational spaces in people's imagination [23].

Marxist scholars such as David Harvey, Edward Soja, and Manuel Castells have further enriched the connotations of the theory of the production of space in their urban and geography studies [31–33]. Harvey combined the theory of capital accumulation with the theory of urbanization, and discussed the mechanism of capital's accumulation, circulation, proliferation and spatialization in the process of urban space production and reproduction [34]. He put forth the belief that capital continues flowing, circulating and proliferating through three circuits in space (Figure 1), in order to alleviate the crisis of over-accumulation and obtain more surplus value [35–37]. First, the primary circuit of capital is mainly based on direct production and consumption, which creates the foundation of capital accumulation. Capital flows into the factories, shopping malls and other spaces to promote the production and consumption of products, and to generate surplus value with which the scale of production can be continuously expanded and capital proliferation can be realized constantly. Subsequently, when overproduced products surpass consumers' demand, and the rate of return on capital falls to unprofitable levels, surplus capital enters the secondary and tertiary circuits. The secondary circuit consists of fixed capital that supports the production and consumption fund which in turn supports consumption. In the field of production, surplus capital flows into fixed capital such as the producer durables (e.g., machines, equipment, etc.) and the associated built environment (e.g., factories, warehouses, roads, ports, etc.). This prolongs the period of return on investment, alleviates the crisis of excess capital, and improves labor productivity through investment. In the consumption field, surplus capital flows into the consumption fund to create consumer durables (e.g., TVs, refrigerators, cars, etc.) and related built environments (e.g., stores, supermarkets, houses, etc.). While delaying the cycle of return on investment and the crisis of over-accumulation, this also promotes the reproduction of labor power (that is, the laborers meet their daily needs through consumption, which helps to maintain and reproduce their capabilities to perform paid work). In the tertiary circuit, capital also flows into fields with a longer period of return on investment, primarily composed of technology and science expenditures supporting the production and social expenditures which support consumption. In the field of production, the state functions drive capital

to invest in scientific research, technology research and development, and relevant built environments (e.g., scientific research centers and buildings) to enhance the innovation capability and long-term competitiveness in production. Within the field of consumption, state functions drive capital to invest in the fields of collective consumption and social expenditures, including education, health, welfare, public security, national defense, and related built environments (e.g., schools, hospitals, parks, welfare institutions, police stations, etc.), in order to promote the reproduction of labor power. It can be seen that capital's three circuits in space promote capital accumulation not only by producing, circulating and consuming commodities in space, but also by continuously creating new spaces and built environments. The continuous and periodic process of capital to create the built environment and physical infrastructure for production, circulation, exchange, and consumption is actually the process of urban space production as well as the process of urbanization [35–37].

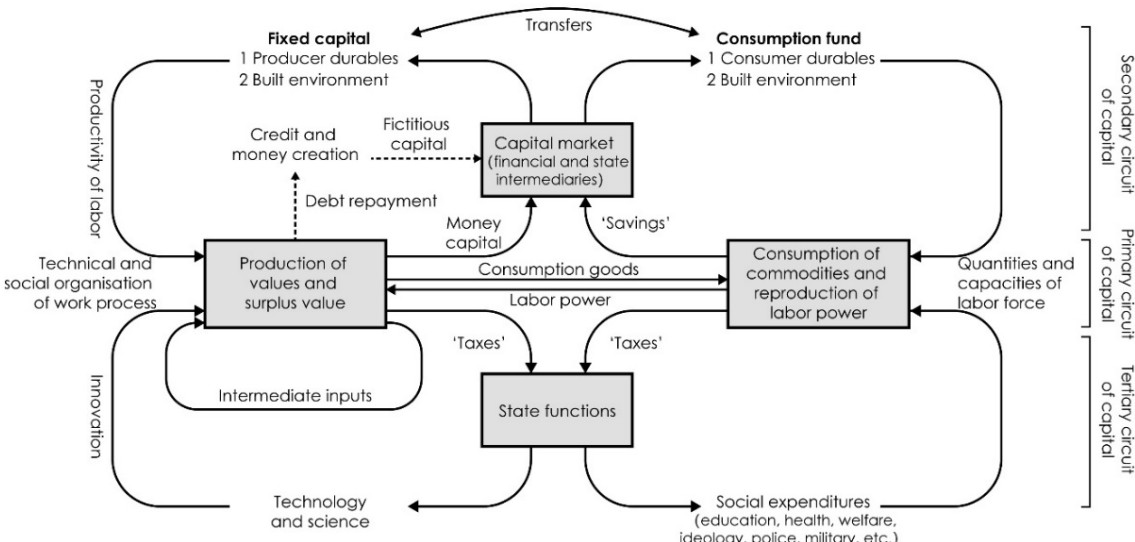

**Figure 1.** The primary, secondary and tertiary circuits of capital in space. Author's illustration based on David Harvey's research.

Harvey used the concept of the "temporal-spatial fix" to explain and summarize the mechanism of the three circuits of capital in absorbing excess capital and labor. The temporal-spatial fix can be understood as the transfer of excess capital to the interrelated temporal and spatial dimensions to temporarily prolong the return cycle of capital and alleviate the crisis of over-accumulation [38]. In the temporal dimension, excess capital is continuously transferred to long-term investment projects or social expenditures. In the spatial dimension, excess capital is constantly being transferred to urban built environments, such as new factories, houses, shops, roads, and even to new towns and new districts around the city, promoting continuous urban growth and expansion. However, Harvey put forward the belief that the effects of the temporal-spatial fix on alleviating the crisis of capitalism are also limited, especially when investment exceeds a certain critical point, at which point obtaining returns will become more difficult. At that time, the exchange value attached to the built environment is bound to depreciate, decrease, or even disappear completely [35]. For example, when a large amount of capital pours into cities and is directed to new built environments, such as excessive or more productive factories and offices, this leads to a depreciation of the exchange value of existing fixed capital, such as the old built environments. However, the old built environments still have different degrees of use value according to their construction year, quality and maintenance level. As a result, these old material environments can be traded continuously in the market as depreciated capital, and after different degrees of renewal in terms of material and function, they become the material carriers that promote the accumulation and proliferation of

new capital. Such a phenomenon, i.e., that the urban built environment continues to experience the process of devaluation, transaction, renewal and appreciation after space production, can be understood as the reproduction of urban space. Space reproduction is often accompanied by the circular flow of capital and the inherent material and economic life cycle of the built environment, showing periodic laws and characteristics. Due to the cyclical depreciation of the built environment as fixed capital, the cyclical input of new capital and space reproduction can be promoted, which curbs the decline in profit margins, accelerates a new round of capital value growth, and reshapes the urban landscapes and built environment.

Space reproduction is also a persistent game and competition for the built environment, between old and new capital as well as the powers, stakeholders and production methods behind it. The old capital produces and shapes the physical space that represents its own image and accumulation logic, and drives the continuous accumulation of capital. Although the built environment faces cyclical depreciation, the process or logic of capital accumulation has been solidified within it, becoming a long-standing spatial barrier that prevents and restricts new capital–representing higher productivity–from reshaping and replacing it. When the new capital and its stakeholders have a greater advantage in the game, they will carry out the "spatial practices" of urban regeneration, such as renovation and reconstruction. Additionally, by planning the "representations of space", or spatial order, which represents their interests, they shape new "representational spaces", and finally realize the reproduction of urban space. As a result, capital promotes space production and the reproduction of cities through continuous circulation, fluctuations, and games in time and space.

### 3.2. Space Reproduction and Urban Regeneration

Some Chinese scholars have attempted to analyze urban regeneration in Chinese and Western cities from the perspective of space reproduction [13,21–23,28,39]. According to the aforementioned perspective, urban regeneration refers to the situation whereby the city government, as well as industrial and commercial capital, pursue the appreciation of spatial assets and the income derived from differential land rent while also realizing the proliferation and accumulation of capital, by the upgrading or replacement of spatial intensity, environmental quality, land use type, geographical location, etc. [13,40,41]. Since the twentieth century, both Chinese and Western cities have been promoting urban regeneration. The forms and emphasis of space reproduction exhibit differences in diverse historical periods and backgrounds.

### 3.2.1. Space Reproduction in Western Cities

Since the twentieth century, Western cities have roughly undergone three different stages of space reproduction [13]. The first stage began in the 1930s. The cities that developed and expanded after the Industrial Revolution carried out large-scale urban regeneration due to the periodic decline of the physical environment and the destruction caused by war. The practice of space reproduction in this stage was led by the government, with the means of demolishing slums and rebuilding houses on a large scale, and with the aim of improving the quality of the urban environment and living. By demolishing the built environment whose use and exchange value have been severely depreciated, and investing capital to produce new, higher-quality housing and other kinds of built environment for consumption, the government achieved the stimulation and supply for collective consumption demand and the proliferation of capital. The second stage began in the 1960s. With the increasing economic prosperity and the growing middle class after World War II, Western cities also faced social crises such as the movements of social democracy and civil rights, as well as large-scale riots, while a series of urban regeneration practices were launched in response to those social problems. Regarding space reproduction in this period, the government introduced a public participation mechanism, using the means of neighborhood restoration and renewal, and aiming to improve the quality of life

of existing communities, ameliorate the quality of social services, and solve social problems. Through consultation with citizens, the government invested capital to improve and upgrade the depreciated and problematic living environment, which not only promoted the appreciation of living space, but also solved social problems to a certain extent, optimized social capital, and ensured the continuous reproduction of labor power. The third phase began in the 1970s and continues to this day. Influenced by the economic crisis, the urban regeneration model under Keynesianism, dominated by the government and supported by public finance, became unsustainable, and began to transform into a market-oriented model under neoliberalism [42]. The form of space reproduction in this stage primarily adopted the public-private partnership mechanism, which was led by the urban growth alliance, composed of the public sector of the government and the private sector of the market. Such space reproduction practices focused on the spatial redevelopment of the city center (e.g., old city center gentrification, old waterfront revival), with the goal of promoting investment and consumption, increasing local taxes, improving environmental quality, enhancing the city's brand image and reputation, and other sustainable development goals. The government was responsible for creating favorable policies and institutional conditions, while the private enterprises, as the main investors and implementers, usually renovated or demolished and rebuilt the depreciated built environment in the city centers, by transferring production places (e.g., old docks and factories) into new consumption and production places (e.g., shopping malls, residences, art galleries, parks, office buildings, etc.), or by upgrading the old residences to commodity housing with higher quality and increased cost. Through the redevelopment and reproduction of space, the re-accumulation and proliferation of excess capital in the built environment was realized.

### 3.2.2. Space Reproduction in Chinese Cities

Since the twentieth century, Chinese cities have also experienced multiple rounds of urban regeneration [2,3,6], which can be roughly broken down into four stages of space reproduction. The first stage spanned from 1949 to 1977. After the founding of New China, the cities that had experienced many years of war became increasingly dilapidated, finding themselves in urgent need of renewal. However, due to the lack of financial resources, the government prioritized investing capital in production areas (e.g., the construction of new industrial zones, etc.); as a result, the funds invested in the renovation of the old city were very limited. Therefore, the practice of space reproduction at this stage was led by the government, using small-scale repairs and maintenance of the old city as mechanisms, focusing on repairing typical shanty towns and dilapidated old houses, and aiming to improve the basic environment and living conditions. In other words, the government reproduced only a few spaces, whose use value had been severely depreciated, in order to maintain the operation of basic urban functions. Against the background of socialist public ownership and the planned economy at that time, land and the buildings it carried were all public assets. The renovated and newly-built houses constituted only a welfare product provided by the government and had no exchange or investment value.

The second stage spanned from 1978 to 1989. This was a transitional period when China was promoting reform and opening up and transforming from a planned economy to a market economy. Urban regeneration was also explored and attempts in this regard were made. The practice of space reproduction at this stage was also led by the government, which used large-scale urban renovation as a means, and aimed to solve the problems of housing shortages and infrastructure insufficiency. The content of space reproduction practice was more diverse, including not only the overall functional adjustment, structural optimization, and beautification of the old cities in Hefei and Shenyang, but also the renewal of the commercial districts in Shanghai and Nanjing, as well as the traditional neighborhoods in Beijing and Suzhou, alongside the "public-private co-construction" of the old city in Guangzhou. The 1988 Constitutional Amendment officially allowed local governments to transfer land-use rights through public auctions, on the premise that urban land ownership belongs to the state. This endowed the land with exchange value and

investment potential, and the government was able to attract market capital and social funds to participate in old city renovation. On the one hand, this alleviated the problem of government fund shortages, and improved housing and infrastructure conditions as well as urban functions. On the other hand, under the socialist system, the above-mentioned amendment also explored the market-oriented circulation mechanism of land, which represents the core element of spatial production.

The third stage spanned from 1990 to 2011. At this stage, China's market economy transformation was further deepened, and China joined the World Trade Organization in 2001 to fully integrate into the global economy. In this context, industrialization and urbanization also entered a period of rapid development, and a large amount of capital flowed into the development and space production of various new cities and new districts [43]. At the same time, with the maturation of the land market, the gradual termination of the welfare housing system, and the continuous advancement of housing commercialization reform, the mobility of major spatial elements (e.g., land and housing) in markets gradually increased, enhancing the attractiveness of market investment, and providing a strong political and economic stimulus for old city regeneration. At the same time, with the reform of the tax-sharing system in 1994 and the "Decision on Deepening Reform and Tightening Land Administration" issued by the State Council in 2004, local governments gradually gained the power to share the profits from land sales and urban renewal, which greatly enhanced their enthusiasm in participating in urban redevelopment practices. Therefore, the practice of space reproduction at this stage was mainly led by local governments and market players (e.g., real estate developers, financing platforms, etc.), forming a Chinese-style "urban growth alliance". By means of large-scale old city renovation and old area redevelopment, the goal of space reproduction was to promote rapid economic growth, upgrade the industrial structure, and enhance the city's brand image. Specifically, local governments and their platform companies usually focused on the early phases of space reproduction, such as site selection, planning and design, land preparation, demolition and resettlement, etc., while market players paid more attention to the later phases, such as land auctions, development and construction, and operation. Local governments often obtained initial financing by selling the land, which was used to pay for the space reproduction costs of public built environments (e.g., land preparation, infrastructure and public facilities). With the capital and technology of domestic and foreign market players, space reproduction practices in depreciated areas (e.g., old industrial areas, old cities and towns, old villages, etc.) usually promoted the appreciation of spatial capital and benefits generation. During this period, the renewal of production space was often associated with industrial structural upgrading ("suppress the second industry and develop the third industry"), the outward relocation of traditional industries from the central city, and the transformation of the industrial space. For example, the World Expo Park project in Shanghai gradually renovated the old waterfront industrial area into a new urban space covering both production and consumption functions (e.g., office, exhibition, culture, commerce, training, residence, leisure, etc.); the 798 Art District project in Beijing transformed the old industrial area into a cultural and creative industry cluster. The renewal of consumer space was mainly reflected in the renewal and value enhancement of old residential areas, urban villages, and old commercial areas. For example, in the Zhongyuan Liangwan Town project in Shanghai, the largest dilapidated urban area at that time was demolished and rebuilt into a new commercial residential area; in Guangzhou and Foshan's "three-old reconstruction" (i.e., old town, old factory and old village), urban villages were demolished and reconstructed into commercial housing of a higher quality and price.

The fourth stage spans from 2012 to the present. In this stage, China's urbanization rate exceeded 50%, and urban development gradually shifted from large-scale incremental construction to high-quality urban development and regeneration. Against the background of increasing emphasis on urban governance modernization and public participation, the space reproduction practices at this stage involve local governments, market players and social groups, while they also formed cooperation modes with differential degrees

and types [2]. The means of space reproduction become more diversified, refined and progressive as a result, including not only the renovation of old industrial areas, shanty towns and old communities, but also ecological restoration and urban repair, community micro-regeneration, and the transformation of creative industrial parks. Under the guidance of the country's "five-in-one" overall strategic layout, the goals of space reproduction also become more diversified. Besides emphasizing economic benefits, the layout also pays more attention to the social and ecological dimensions of sustainable development goals, such as people-oriented, social equity, happiness, livability, ecological preservation and low carbon values. The specific practice is similar to the third stage. Local governments and their platform companies mainly invest in land preparation, public facilities, and environmental quality improvement. The said governments obtain economic benefits through land transfer and future taxation, and pursue the goals of improving the space value, the city image, the environmental quality and the life quality of citizens. Market players mainly invest in the renovation, development and operation of real estate, and obtain economic benefits by selling, leasing and operating properties. Relevant social groups seek a more reasonable redistribution of renewal interests (e.g., economic compensation, improvement of living environment) through public participation and social co-governance.

## 4. Researching Relevant Theoretical Framework

From the perspective of space reproduction, the urban regeneration of Chinese and Western cities since the twentieth century has generally focused on urban spaces whose use value or exchange value has depreciated, and carries out different degrees of spatial reproduction practices ranging from repairs and small-scale renovation, to large-scale demolition and reconstruction in order to enhance the value of space. The improved space value includes not only the use value related to the environmental quality, and the exchange value of the built environment as fixed capital and commodities, but also the brand value of the spatial image, the cultural value of historical space, the social value of public space, and the ecological value of natural and green space, etc. Hence, the values, interests and goals pursued by space reproduction have exhibited diversified and comprehensive characteristics, while the stakeholders, cooperation mechanisms, regeneration objects, and benefit distribution have also become complicated. Therefore, it is necessary to build a systematic and integrated theoretical framework to provide theoretical support for a more accurate and comprehensive understanding and analysis of urban regeneration practices.

Some scholars have discussed the theoretical framework of urban regeneration from the perspective of space reproduction [22,23,44–46]. These discussions refer to diverse types of urban space reproduction, for instance, industrial space, cultural space, residential space and commercial space (Figure 2), and mainly involve the stakeholders, game relationships, and operating mechanisms of space reproduction.

### 4.1. Framework on Industrial Space Reproduction

For example, Li Zhigang et al. studied the space reproduction process of the "Chu Milky Street" project in Wuhan from an old factory area to a commercial district, and established an analytical framework composed of the following three types of stakeholders: the government, the market, and society [22]. Among them, the government (Hubei province, Wuhan city, and Wuchang district), as the project initiator and interest coordinator, occupies a dominant position, and is responsible for land preparation and management (land acquisition and storage, land transfer, demolition and resettlement), planning and design, project supervision, etc. Finally, the government realizes the improvement of the project area in terms of taxation, employment, urban quality, ecological environment, cultural brand and other aspects. For the market players, namely the original land holder (Wuzhong Group) and the developer (Wanda Group), the former increases the land area of the new factory, the upgrading funds for equipment and technology, and the competitiveness of the enterprise, by returning the land and coordinating the relocation. The latter obtains the property rental and sales profits and brand reputation by being responsible for the

investment, design, execution and management of the project. The social subjects include the original residents (Wu Zhong's staff) and related citizens and tourists. Among them, the original residents obtain resettlement houses with higher prices, as well as better living environments and service facilities, through cooperative relocation. Citizens and tourists become the consumer groups of the new space.

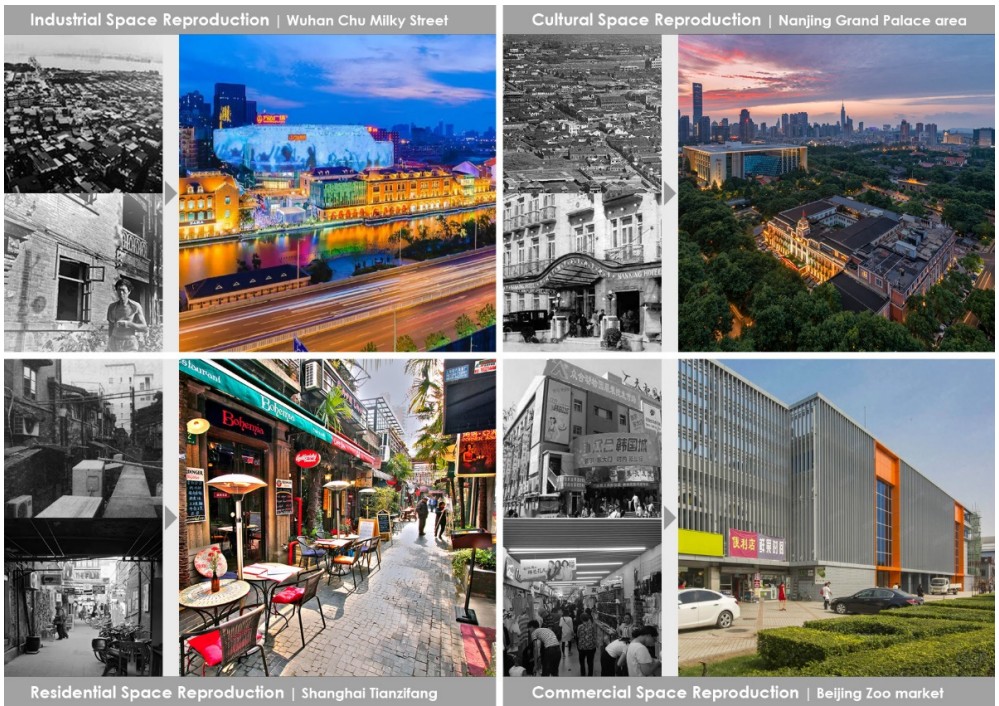

**Figure 2.** Examples of industrial, cultural, residential and commercial space reproduction. Author's illustration based on online images.

### 4.2. Framework on Cultural Space Reproduction

Sun Shijie et al. constructed an analytical framework for the reproduction of the old city space, and analyzed the process of cultural space reproduction in the Nanjing Grand Palace area [23]. In this analytical framework, social groups include stakeholders such as government officials, developers, land occupiers and native residents, as well as technical service providers such as planners and architects, and space users and consumers such as operators, citizens, and tourists; these stakeholders are not only the subjects and key actors in the process of space reproduction, as well as the owners and users of power and capital, but also the seekers and gainers of interests. Interests, such as economic or non-economic, public or private, material or spiritual, are the goals pursued by various social groups using the capital and power they control. During the process of space production, capital, such as finance, land, technology, cultural and social capital, etc., acts as an instrumental factor that promotes accumulation and proliferation, while power, such as political, economic and social power, is manifested as the dominance, profit and discourse power obtained by stakeholders through competition and games. In the case of the Grand Palace, due to the significant cultural value of the area, the government is leading a multi-party cooperation mode of development. This area is reproduced as a cultural space that brings together libraries, art galleries, museums and traditional commercial streets. Economic capital and cultural capital are embedded into the material space carriers, such as exhibition halls and commercial streets, which can be spatialized, solidified and accumulated, while the consumption of cultural space and the proliferation of capital can be realized by citizens and tourists visiting the exhibition halls and commercial streets.

### 4.3. Framework on Residential Space Reproduction

In addition, Zhou Yu studied the renovation process of Shanghai Tianzifang from a traditional lilong dwelling to a cultural, artistic and creative economy area, and constructed a research framework for space reproduction [44]. The research framework reflects the power relations of the following three types of stakeholders in Tianzifang's space reproduction: the government (Administrator, local committee, sub-district office), market enterprises (businessmen, agencies, artists, investors, operators), and the public (landlords, left-behind residents, tourists, and the media). Under the domination of consumer culture, various stakeholders have formed a network of power, capital and culture around the use value, cultural value, and especially the brand and symbolic value, of the Tianzifang Lane space. These stakeholders also rely on the production and consumption of symbolic value to accelerate the circulation of capital.

### 4.4. Framework on Commercial Space Reproduction

Furthermore, Shen Haojing et al. constructed an analytical framework including government, market, society, and culture based on the theory of space production, and explained the reproduction process of the transformation of Beijing's old commercial market into an industrial space of technological innovation [45]. This framework adopts the analytical perspective of "power-capital-daily life", and explains the interrelationships between key factors formed in the space reproduction process, including "government" as the main controller of power, "market" as the main owner of capital, and "public" and "culture" as the main components of daily life. In the case study, the government uses power to guide the upgrading of industry, and the market invests new capital to pursue profit maximization. The two together promote the replacement of traditional low-value-added production space with high-value-added and high-profit space, while the participation of the public in this process is fairly limited. At the same time, the culture accumulated in the space, especially after the formation of cultural symbols and brands, also plays an important role in enhancing the capital value of the space as well as the cultural identity of society.

In general, the current analytical or theoretical frameworks based on the perspective of space reproduction include the following three categories of stakeholders: the government, the market, and society. Various stakeholders use the numerous types of powers and capital under their control to compete, cooperate and play games in the process of space reproduction, in order to pursue and obtain their own interests. On the basis of summarizing and integrating the existing knowledge, there is still some content that can be further optimized and supplemented in order to form a more systematic and comprehensive theoretical framework. First, the current theoretical framework usually treats space or space production as a single element and explores its relationship with various stakeholders. However, some key concepts and mechanisms in the theory of the production of space constructed by Lefebvre and Harvey are not included and expressed, such as spatial practice, representations of space, representational spaces, and capital circulation. Secondly, the current research on space reproduction practice and the theoretical framework mainly focuses on the roles and functions of the government and the market, while the cognition and analysis of the roles, functions, and importance of social subjects are relatively weak. In today's world of pursuing sustainable economic, social, and environmental development, and considering China's emphasis on people-oriented governance modernization, further discussion and research are needed on social subjects and social dimensions. Thirdly, most frameworks are established on the basis of summarizing or guiding a single specific case, and their comprehensiveness and generality still need to be enhanced. Combining existing regeneration practice and research, we can further summarize and classify the composition of each subject and element in the framework, as well as the different cooperation mechanisms between them.

## 5. Constructing the Theoretical Framework of Urban Regeneration

This part will build a systematic and comprehensive urban regeneration theoretical framework from the perspective of space reproduction on the basis of the research concerning the theory of the production of space, space reproduction practice, and related theoretical frameworks, combined with other related theoretical works (Figure 3).

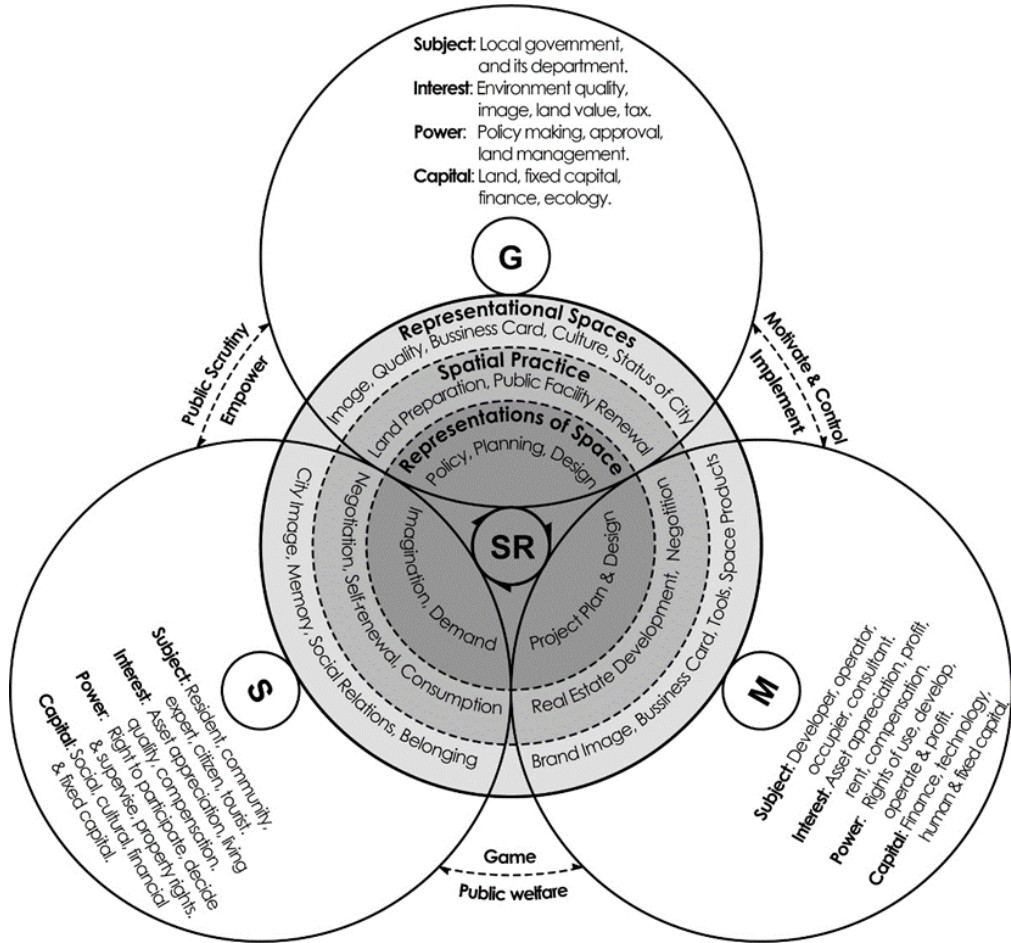

**Figure 3.** The theoretical framework of urban regeneration from the perspective of space reproduction. SR represents Space Reproduction, and G, M and S represent the Government, the Market and the Society, respectively.

The framework contains the four following core elements: the government, the market, society, and space reproduction. The first three elements are the three types of stakeholders in the process of space reproduction, and the last element reflects the internal mechanism and process of space reproduction. The overlap between the first three elements and space reproduction reflects the planning, practice and focus of various stakeholders in the process of space reproduction. At the same time, the three types of stakeholders are interrelated, and different types of co-opetition and game relationships have been formed in practice.

### 5.1. The Government

The government, especially local governments at the provincial and urban levels, and their related departments, usually plays an important role as initiators, leaders and coordinators in the process of space reproduction. Local governments use their functions and powers in policy formulation, land management, planning and design approval, and project supervision to promote a series of tasks, such as setting goals, clarifying requirements, selecting subjects, promoting implementation, safeguarding public interests, and supervising operations during urban regeneration [47]. The main types of capital used

by local governments for space reproduction include land capital, fixed capital, financial capital and ecological capital. Among them, land capital is the core capital of space reproduction, and is divided into the following two categories under China's public land ownership system: state-owned land and collective land. Urban regeneration normally involves developed state-owned land located in urban areas (e.g., urban construction land). Under the authorization of the central government (as the sole representative of state-owned land), local governments can dispose of state-owned land (e.g., land acquisition and storage and transfer), and exercise the right to benefit from it. In addition, local governments can also expropriate developed collective land in urban and suburban areas (e.g., urban villages) due to public interests, and convert it into state-owned construction land for urban redevelopment and space reproduction. Fixed capital mainly refers to the dilapidated or depreciated public built environment occupied by local governments and involved in space reproduction, including infrastructure such as roads and docks, and public service facilities such as schools, hospitals, museums, and parks. Financial capital involves funds used by local governments for urban regeneration, including central policy funds and special bonds, local financial funds (including land transfer fees), and local government bonds. Ecological capital primarily includes ecological resources and an ecological environment that can bring about economic, social and ecological benefits in urban regeneration, such as rivers, mountains, green spaces, etc. As an important public form of capital and resource of a city, ecological capital is increasingly valued by governments in the era of ecological civilization. For example, in the practice of "ecological restoration and urban repair", the value of ecological capital is restored and enhanced through the ecological restoration of urban ecological elements and systems. Overall, local governments use their power to invest all kinds of capital in the reproduction of urban space in order to seek the growth of interests. These interests are mainly public and economic interests, involving the improvement of the urban living environment, ecological environment amelioration, urban image enhancement, land value increase, and corporate tax increase.

### 5.2. The Market

The market generally refers to the relevant subjects and enterprises in the market economy, which often play an important role as responders, investors and implementers in the process of space reproduction. These market players mainly include developers, investors, operators, space occupiers and consulting companies. Among them, real estate developers are the main executors of space reproduction, responsible for the renovation and redevelopment of residential buildings, shopping malls, factories, warehouses, office buildings and other real estate, as well as for supporting ground roads and underground infrastructure and other real estate, in order to create higher-value spatial products. A developer can also be an investor or operator at the same time. Investors are market entities that provide funds for the whole process of demolition, renovation, construction, operation, etc., including not only developers and strategic investors, but also policy banks (e.g., China Development Bank), commercial financial institutions, such as banks (e.g., China Construction Bank), trust companies, security companies, and fund companies. Operators refer to enterprises that provide operation management and other services for the renovated spatial products, including operators of shopping malls, industrial parks, long-term rental apartments, and property companies in commercial housing communities, etc. Space occupiers, as key stakeholders, include both the occupiers of the original land or building space and new space occupiers after renewal, such as newly-settled enterprises. Consulting enterprises are those enterprises and institutions that provide professional technical solutions and consulting services for the whole process of space reproduction, including planning and design companies, architectural design companies, law firms, real estate consulting companies, etc. The main capital controlled by the market players includes financial capital, technological capital, human capital and fixed capital. Among them, financial capital is the core capital controlled by the market to promote space reproduction and capital proliferation, including

self-owned funds, bank loans (e.g., CDB, CCB urban regeneration loans), non-standard financing (e.g., trust, private financing), bonds funds (e.g., urban investment bonds), urban regeneration funds (e.g., Shanghai Urban regeneration Guidance Fund), REITs real estate investment trust funds (e.g., urban regeneration REITs), etc. Technological capital, human capital, and fixed capital involve various technical processes, professionals, equipment, and real estate (e.g., factories and office buildings held by space occupiers), that various market enterprises direct towards space reproduction. Numerous types of market players have used their capital and acquired spatial development rights of development, operation, use, and profit to promote capital investment, planning and consulting, scheme studies, development and operation; these players have obtained various benefits, such as the asset appreciation of real estate, an increase in capital profits, an increase in space rent, compensation for demolition or renovation, improvement of the operating environment, and an increase in the brand and market value of the enterprise, etc. It is worth noting that some state-owned enterprises supervised by the central or local governments, such as state-owned developers, local investment and financing or development platform companies, often take into account both economic and public interests, participating in public renewal projects with long payback periods. Therefore, the attributes and roles of state-owned enterprises that assist the government in providing urban public goods and services are more akin to those of the government.

*5.3. Society*

The term society mainly refers to the social public with relevant interests in the process of space reproduction, who usually play the role of cooperators and participants, and include local residents, community organizations, surrounding residents, experts and scholars, as well as other citizens, migrants and tourists. Among them, local residents refer to the aborigines or villagers living on the renewed land. They are the most important stakeholders in the regeneration, and the main users and consumers of residential and living spaces such as houses and communities. Community organizations, e.g., homeowners' councils, are the main representatives of the rights and demands of local residents. The surrounding residents of the renewal area are also influenced by external effects such as environmental improvement and increases in house prices brought about by the renewal, and become stakeholders of relevance. Experts and scholars participate in the space reproduction mainly by means of expert review and public participation in regeneration planning. Other citizens, migrants, and tourists are often involved in the role of users and consumers of the regenerated urban space, i.e., the consumers of new commercial centers, commercial housing, art galleries or urban parks. Compared with the government and the market, the public has less power and capital, including the right to participate, to know, to make decisions and to supervise, and the right to occupy, use, and benefit from their own property, as well as social capital, cultural capital [48], fixed capital, and financial capital. Regarding capital, social capital here refers to the social relationship network and human network resources formed by local residents around their living space. These kinds of social capital are conducive to promoting mutual understanding, mutual trust, mutual cooperation, identity, social norms and common values, etc., thereby contributing to the more harmonious operation and development of society. Cultural capital involves the tangible historical and cultural relics, resources, features and textures of the renewed space, as well as the intangible cultural traditions, activities and brands, which together maintain and illuminate the spatial image of the urban historical space in the public's collective memory. Fixed capital primarily refers to residents' real estate, which has both consumption and investment attributes, and can create economic benefits in terms of rent or asset appreciation after renewal. Financial capital relates to the financial capital invested by the public in self-renewal, such as deposits and loans. Using their power and capital, the social public participate in the formulation of renewal plans, project implementation and supervision in order to obtain real estate value enhancement (or house price increase),

compensation for demolition, and the improvement of housing conditions, service facilities and living environment.

*5.4. Space Reproduction*

Space reproduction is actually a circular game and competition between old and new capital as well as the powers and subjects behind them vying for spaces with higher investment value. There exist three major elements in Lefebvre's theoretical framework of space production, namely representations of space, spatial practice, and representational spaces, which also implies the internal mechanisms of the planning, developing and experiencing of urban space. The government, the market, and society have their own emphasis on the planning, practice, and experience of space reproduction.

5.4.1. Representations of Space

Concerning the representations of space, the government usually plans the "representations of space" or spatial order, which reflects their interests by formulating urban regeneration policies and organizing renewal plans and designs. For example, the government guides the main objects, goals, principles, methods and strategies of space reproduction by formulating policies such as "three-old reconstruction" and "ecological restoration and urban repair". In addition, by organizing the preparation of urban planning and design, the government controls the consumption pattern, land-use layout, industrial function, land ownership, road network system, spatial form, landscape features, etc. Enterprises in the market usually lead, or participate in, the specific project planning and product design based on their own developmental strategies and profit models. Their plans and designs are most suitable for the economic interests of enterprises, and represent the higher-profit spatial products in the future. For example, a regenerated space planned by a developer, investor or operator actually represents a spatial product with a higher, faster, and longer-lasting return on investment. The public, mainly through the public participation mechanism of planning, put forward proposals and demands for the future space that represents their vital interests. For example, residents can put forward demands and ideas for space reproduction for future residential spaces, such as housing, public squares, green spaces, parking spaces, etc. In general, the government, the market, and society as three types of stakeholders, exhibit a kind of large-to-small and macro-to-micro characteristics in terms of the degree and scope of their influence on the "representations of space".

5.4.2. Spatial Practice

Secondly, in terms of spatial practice, the local government or the local state-owned platform company is mainly responsible for the preliminary urban land preparation or land development, including land acquisition, demolition, resettlement, compensation, financing, municipal facilities construction, land reserve and transfer, as well as the renewal work of existing public facilities and environments, such as the renovation of municipal and public service facilities, and ecological restoration. Market enterprises are mainly engaged in real estate (such as residences, shops, factories, warehouses, office buildings, etc.) investment and financing, renovation and development, product rental and sales, and operation services, as well as the construction of affordable housing and public service facilities. Among them, state-owned enterprises usually cooperate more deeply with the government to participate in projects associated with strong policy preference, high investment, public welfare, and people's livelihood, while privately- and foreign-funded enterprises pay more attention to those projects with higher return on investment and larger profit margins. The enterprises as original space occupiers mainly participate in the spatial practice by negotiating renovation or cooperating with relocation. Additionally, the spatial practices of the public involve negotiating renovation or relocation, resettlement and compensation, self-renewal practice, and the use and consumption of regenerated space by residents, citizens and tourists. In general, the three types of stakeholders have different

emphases in "spatial practice". The government focuses more on the reproduction of land space and public space, the market focuses on the reproduction of architectural space and commercialized space, and society mainly participates in the reproduction of residential architectural space and the consumption of renewal space.

5.4.3. Representational Spaces

Finally, the representational spaces reflect the symbolic, imaginary, emotional, and historical and cultural intentional spaces of various stakeholders superimposed on the renewal material space. For the government and officials, the reproduced space not only has material and functional attributes, but also symbolizes the image and quality of the city, the city's business card, the strength and status of the city, and the city's history and culture. For example, the Huangpu riverside space, renovated under the leadership of the Shanghai municipal government, has become a "world-class urban meeting room" and a city business card for the government to display and represent the image of Shanghai. For market enterprises, the reproduction space symbolizes the brand image of the enterprise, the business card, the production tools that create value, and the consumable spatial products. For example, from the perspective of developers, the renovated or redeveloped building spaces, such as long-term rental apartments, commercial buildings, and commercial centers, not only represent the spatial products that can be rented, sold, and consumed, but also symbolize, to a certain extent, the brand image of the company. The regenerated workshops and office buildings become the production tools in which the territorial enterprises can use to create value more efficiently or lastingly. For the public, the regenerated material space still superimposes and represents people's cognition and intention of the city (e.g., city centers, landmark buildings, main streets, etc.), as well as the memory of historical culture (e.g., traditional buildings, historical neighborhoods, custom activities, etc.), perception of social relations (e.g., neighborhood space, unit compound, clan settlement, etc.), and personal emotional belonging (e.g., childhood home, urban space with a sense of belonging or emotional connection, etc.). The greater the change in the urban material space, the greater the change in the representational spaces for the public, and the greater the loss of social and cultural capital, and emotional memory accumulated in space. In general, the "representational spaces" of three types of stakeholders also have differing emphasis. The government focuses more on the public symbolized image of renewal space, while the market focuses more on the production and consumption attributes superimposed on the renewal space, and the public pays more attention to the social, cultural and emotional connotations of spaces.

*5.5. Co-opetition between Stakeholders*

The three types of stakeholders are interrelated, and the space reproduction practices of Chinese and Western cities in different periods have formed several main co-opetition models. The first is the government-led model, in which the government uses its power and capital to dominate and promote the practice of space reproduction. This model has appeared in post-war Western cities and Chinese cities. The urban material environment is in urgent need of renewal, and the capital strength of the market and social subjects is still weak, thus meaning that the government has to lead urban regeneration. The second is the model of government domination and public participation. This model appeared in Western cities in the 1960s. Cities faced not only physical spatial problems, but also social problems such as social unrest and the intensification of contradictions. Therefore, with enhancing capital strength and awareness of rights, the social subjects, as key stakeholders of spatial and social problems, participate in government-led urban regeneration practices. The third is the growth alliance model formed by the government and the market. This pattern emerged in Western cities after the 1970s, and in Chinese cities after the 1990s. In addition to the problem of spatial decay, cities were also faced with economic problems such as economic crisis and government financial difficulties. Therefore, the government and market players have formed an alliance to jointly promote urban regeneration based on the

consensus of using the financial capital from the market and sharing the benefits of renewal. However, under this model, the government and the market tend to place more emphasis on political performance and economic benefits, and pay attention to the exchange value of urban space, but ignore the actual use value and reasonable demands of the aborigines for space to some extent. Indeed, this results in the emergence of spatial injustice, such as the injustice of the mechanisms for expressing demands and the distribution of spatial benefits, all of which normally force residents to move to the urban fringes and suburbs [49]. At the same time, the social capital, cultural capital and emotional memory attached to the built environment that the public can enjoy, are easily ignored and destroyed by the economic interest-oriented urban growth alliance in the process of space reproduction. For instance, demolition and relocation bring about the collapse and loss of traditional social network, the destruction of historical and cultural relics, the disappearance of traditional spatial features and patterns, and the loss of place memory and sense of belonging, etc.

The interests of society and the social dimension of development have received increasing attention in recent decades. Since the United Nations published the Brundtland Report in 1987, the concepts and theories of sustainable development, including the three dimensions of economy, environment and society, have been widely recognized and gradually become the standard to guide sustainable urban regeneration. Among them, social sustainability addresses the key qualities and goals of a society's long-term development, and covers numerous scientific and policy topics, including basic needs, well-being, social justice, social inclusion, social capital, public participation, employment, income and security [50]. Well-being and social justice are the two crucial themes. Well-being, as the ultimate goal of human behavior, can be roughly understood as a good life. The theory of social production functions proposed by Siegwart Lindenberg integrates well-being and various related goals, needs, activities, and resources into a hierarchical theoretical framework [51]. According to this theory, human beings improve their ultimate well-being by continually optimizing two universal goals of physical well-being and social well-being. These two universal goals can be achieved by meeting the lower-level five instrumental goals or basic needs, including two types of physical needs–namely comfort and stimulation–as well as the following three types of social needs: status, behavioral confirmation and affection. These five basic needs can then be met by the lower-level activities and the resources at the bottom. Therefore, the resources at the bottom can be understood as the key basis and raw materials for well-being production layer by layer, which include funds, housing, living environment, food, and health for physical needs satisfaction, as well as education, talents, social networks, cultural practices, and relatives for social needs fulfillment [52]. In addition, social justice, a key issue in John Rawls's theory of justice, emphasizes the need for a fair and just distribution of resources, powers, and opportunities that promote or produce well-being. Combined with the theory of spatial production, the labor force that promotes production activities in most cities also needs to own and consume various resources under the principle of fairness to meet the physical and social needs of daily life, so as to enhance happiness and maintain work capability, while also promoting the reproduction of the labor force and realizing the successful circulation of capital. When space reproduction undermines the happiness foundation of the labor, this may lead to lower labor productivity, or even social unrest, stagnation of production, and the blockage of capital circulation. Therefore, in the context of China's emphasis on people-oriented and happiness, the public, as the main stakeholder in space reproduction, need further attention; especially, the important resources used by the public to create well-being and happiness (e.g., housing, living environment, financial resources, social networks, cultural customs, etc.) need to be given more attention and protection, and the principle of fairness and justice should also be upheld in the modification and redistribution of related resources, as well as the distribution of renewal benefits.

The government's emphasis on social interests and governance modernization, coupled with the improvement of the public's awareness of rights, personal wealth, and game-playing ability, has led China to explore and study the fourth model of space re-

production in recent years: the multi-stakeholder co-governance model of government, market and society. This model focuses on the collaborative participation of multiple subjects in decision making and implementation, as well as the multi-party sharing of new benefits after the regeneration. Under this model, the government has moved from "multiple-departments management" to "coordinated governance", and directs more effort towards coordinating the demands and wishes of all parties, empowering the public to participate through institutional design, using policies and regulations to motivate and constrain market players, and coordinating public and private interest and interests distribution. Indeed, market enterprises have changed from "seeking profit only" to "taking into account public welfare", not only to ensure reasonable profitability of investment implementation, but also to rationally share the value-added benefits of land and space, and to take into account more public interests; the public have moved from "expressing demands" to "in-depth participation", and participate in more in-depth interest-related processes, such as regeneration plan evaluation, implementation, public supervision, compensation or game negotiation of benefits [47,49]. Ultimately, this promotes the realization of the multidimensional goals and interests of the government, the market and society.

## 6. Discussion

Due to the wide range of disciplines and content involved in urban regeneration, there remains a lack of systematic and comprehensive theoretical frameworks to lay a theoretical foundation for academic research and provide guidance for renewal practice. Therefore, based on a mixed literature review approach, this paper employs the theory of the production of space as the interdisciplinary basis to integrate related theoretical components [22,23,44–46], and constructs a systematic and comprehensive theoretical framework for urban regeneration from the perspective of space reproduction. The proposed theoretical framework includes four core elements, namely the government, the market, the society and space reproduction. The subject, power, capital and interest of three types of stakeholders (government, market, and society), as well as their differential emphases and various co-petition relations in the reproduction mechanism are suggested.

This paper makes certain new theoretical explorations and contributions on the basis of previous research. First of all, by studying and integrating the theory of the production of space, sustainable development, social production function, social justice, and many other related theoretical and empirical studies, this paper constructs a more systematic, comprehensive and universal theoretical framework of urban regeneration. The framework can be used as a research basis for further in-depth discussions or feedback corrections in related theoretical or empirical research. Secondly, this paper incorporates certain core concepts and mechanisms derived from the theory of the production of space into the theoretical framework, including the representations of space, space practice, representational spaces, and the spatialization and circulation of capital, etc. The differential emphases of various stakeholders in the space reproduction process are discussed, and the understanding and cognition of the urban regeneration mechanism are enriched and expanded. Thirdly, regarding the construction of theoretical framework, the current paper strengthens the importance of social subjects and social dimensions. With the help of a series of related theories, this paper expounds that the protection or fair distribution of important resources or new benefits that create people's well-being is not only conducive to the realization of people-oriented sustainable renewal, but also conducive to continuous and stable space reproduction and capital circulation.

Nevertheless, as the proposed theoretical framework in this paper is primarily based on relevant theoretical literature, its application value for supporting the top-level policy making of urban regeneration and guiding city-level regenerative urban planning and project-level regenerative practices still remains unknown and needs further explorations. Moreover, as the cities in contemporary China continue to face a changing and complex developmental circumstance both domestically and internationally, the validity of this

theoretical framework in explaining urban regeneration mechanism in future still remain unknown and requires further studies to investigate more changing and dynamic factors.

To overcome the above mentioned limitations, future research can be conducted to examine and improve the application value of this theoretical framework. By applying this framework, future research could attempt to address several more practical questions in the space reproduction process regarding "Where, What, Who, How, and When". "Where" involves the site selection issues of investment and renewal. Combining the interests, the bottom line requirements, and the "representational spaces" of the three types of stakeholders, a target system can be formulated and further decomposed into urban regeneration assessment indicators to assess and identify areas with renewal potential and value. "What" refers to the planned functions and spatial form. On the one hand, this refers to what functions and physical form can promote the content of "representations of space" such as capital proliferation, asset appreciation, tax increase, housing improvement, and environmental improvement, while, on the other hand, it also involves "spatial practice" content such as specific land preparation, real estate investment, construction, and operation. "Who" refers to the stakeholders in charge of the implementation. One can refer to the composition of the three types of stakeholders, and combine the relevant subjects and content of the regeneration space to clarify the specific stakeholders. "How" involves the division of labor and distribution of interests. One can refer to the content of space reproduction that each stakeholder focuses on, as well as the inter-subject relationship, to build a multi-coordinated co-governance model and an interest negotiation mechanism. Finally, "when" refers to the investment timing and regeneration timing. Choosing good investment timing and regeneration timing will mean that each subject can reap better benefits, capital can flow and circulate better within the primary or secondary land development, and the spillover effect of space appreciation can better increase the prices of surrounding lands and housings.

Further research can also investigate the changing and dynamic developmental circumstances of Chinese cities in future and their influences on the space reproduction mechanism. Since 2012, China's economy has moved from a stage of external circulation, after the reform and opening up, to a period of international and domestic dual circulation dominated by internal circulation. At the same time, urban development has also moved from the stage of rapid development and expansion to the stage of urban regeneration, which emphasizes connotative high-quality development. It can be seen that the main circulation field of capital is shifting from abroad to domestic, and from urban space production to urban space reproduction. On the one hand, the state is guiding capital into the new economic fields that determine future competitiveness, such as new infrastructure, new energy, digital economy, and high-end manufacturing, as well as the production of related spaces. On the other hand, more capital is being invested in the reproduction of urban space, especially in those urban agglomerations and metropolitan areas with more developmental momentum, higher spatial efficiency, and higher return on investment, thus promoting a new round of higher-quality space reproduction with the goal of higher efficiency. In the future, the reproduction of urban space will continue, similarly to the metabolism mechanism of the human body, constantly replacing old cells with new cells; new capital and power will continue to replace the depreciated and declining old capital and power, building a spatial landscape that represents their image and interests. From a broader perspective, space reproduction actually reflects the competition and reshaping of land space by all the new capital with higher profits and all the new subjects with greater power. Such mechanism has promoted the reshaping of natural space into agriculture, the replacement of agricultural space by industrial space, the transformation of industrial space to science and technology space, and the regeneration of consumer space (e.g., residential, business and leisure space, etc.) into higher-profit or higher-quality consumer space, etc. All these changes and replacements reflect the functioning of capital under the auspices of power, constantly circulating, fluctuating, and competing with each other in time and space.

Hence, it would be interesting and also challenging for future research to continuously reveal the underlying nature and rationales of space reproduction in urban China.

## 7. Conclusions

This paper aims to construct a systematic and comprehensive theoretical framework for urban regeneration from the perspective of space reproduction, by using the theory of the production of space as the interdisciplinary basis and integrating related theoretical components.

A more systematic, comprehensive and universal theoretical framework of urban regeneration has been proposed in this paper, which includes four core elements, namely the government, the market, society and space reproduction. The first three elements represent the most important types of stakeholders in urban regeneration, and the last element reflects the content and process of space reproduction. This theoretical framework further summarizes the specific subject, power, capital and interest of the three different types of stakeholders in the process of space reproduction.

Additionally, some core concepts and mechanisms of space production have been incorporated into the theoretical framework of this paper, such as the representations of space, space practice, representational spaces, the spatialization and circulation of capital, etc. It is found that the three types of stakeholders have differential emphases within the space reproduction process. Concerning the representations of space, the influence of the government, market and society vary differently from large to small and from macro to micro, in terms of the influential size and scope, respectively. With regard to spatial practice, the government, market and society focus, respectively, on the reproduction of land and public space, architecture and commercialized space, as well as residential building and consumer space. Regarding the representational spaces, the government, market and society focus, respectively, on the public image, production and consumption attributes, as well as social, cultural and emotional connotations superimposed on the space.

Furthermore, the importance of social subjects and social dimensions of the urban regeneration framework has been strengthened. Despite the various co-petition models formed among stakeholders in space reproduction practice, such as government-led mode and the government-market growth alliance model, the attention paid to social subjects and their interests seems relatively limited. Different models actually reflect the ability of three types of stakeholders to control power and capital, manipulate space reproduction and obtain their own interests in different eras. Against the background of the government's increasing emphasis on social interests and governance modernization, and the public's continuous improvement of rights awareness and game ability, increasing attention is now being paid to a multi-stakeholder co-governance regeneration model that promotes the collaborative participation of multiple subjects in decision making and implementation, as well as in sharing newly reproduced benefits in urban China.

**Funding:** This research was supported by the joint funding from National Natural Science Foundation of China (72061137072) and Dutch Research Council (482.19.607).

**Informed Consent Statement:** Not applicable.

**Data Availability Statement:** Not applicable.

**Conflicts of Interest:** The authors declare no conflict of interest.

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
