# Peer review of "Space Reproduction in Urban China: Toward a Theoretical Framework of Urban Regeneration"

_land, doi:10.3390/land11101704_

Round 1

Reviewer 1 Report

This paper Constructed a systematic and comprehensive theoretical framework of urban regeneration from the perspective of space reproduction. It is an interesting and meaningful topic. A few minor suggestions for the author.

The author introduced very detailed evolutions (or developments) of the concepts of space production, space reproduction, and urban regeneration. However, for me, the relations (or connections) among them still seems a little weak or not enough. Maybe a few sentences for transition are necessary.

The author constructed the theoretical framework of urban regeneration based on the previous studies. If the author can offer a graphic for the process of theoretical framework development (I mean the section 3.1), the readers can understand it easily.

Maybe the author has already presented all contributions and results in the section of Discussion. I still recommend the author to add a short conclusion for the integrality of a research paper.

Reviewer 2 Report

This paper described a theoretical framework for urban generation. The work is literature based, and appeared to cover a lot of ground on an interesting topic. My main suggestions are to explain how literature was identified during the review process (i.e., inclusion and exclusion criteria,) since this was not mentioned. Further, I would strongly recommend that the structure for this paper is improved. Information presented in this work was interesting, but is quite densely compacted, without the use of any tables and only one figure image to supplement the text. Some stretches of the paper also are not complemented with any section headers. Providing more diverse sources (e.g., photographs and statistical data) would, in my view, help enrich the information in this paper, and help communicate more important information across to the readers. These comments have been documented, in more detail, in my suggestions below, which I hope the author finds useful when reviewing their work.

#1: P1, L26: Please consider expanding upon “opening up”, since this is not particularly clear. I believe this might be in reference to local or national activities being scaled to an international level, but it is not particularly clear from reading only the opening sentence.

#2: P2, L69-89: The rational surrounding the need for a theoretical framework is not easy to follow. If my understanding is correct, top-level design refers to the main conceptual ideas underlying urban regeneration, which can then be filtered down into finer design elements (e.g., grass-roots, and core ideas.) Providing this accurately reflects the author’s intended message, please consider adding perhaps a few examples, or even a concept figure image, that helps readers better understand these concepts and how they relate to each other. There is only one figure image to support this paper, so adding more illustrative information might help communicate across certain information the author’s wish to convey to the reader.

#3: P2, L88 to P3, L101: Since this paper was developed from reviewed literature, it would have been helpful to mention how the review process was generally conducted (e.g., keyword searches, search engines, or thematically) to identify the sources of information. The author mentioned systematic on L88, and I believed the review process also followed this style. When reading further, I noticed that literature was leveraged around other high-level concepts (e.g., space production and reproduction.) The paper also appeared to be centered on a case study country, so the review process might have mixed review style elements from different approaches. There are some similar theoretical design frameworks, which are focused on other areas, available that also incorporate similar elements to build their reviews (e.g., Ko et al., 2022; Cabanek et al., 2020; Berke et al., 2014,) which outline these attributes, to their review process, more explicit. I think that the approach used in this study has some overlapping similarities to some of the above, and outlining these to the readers could help connect this to information that followed the opening sections, since section 2.2 also has a difficult narrative. Reviewed information here appeared more along a chronological timeline. Although it is useful to refer to historical endeavors to trace back fundamental source information, its practical implications to the framework were not made overly clear. This issue again appeared to be tied back to how the review was conducted, and so I would suggest this is explained in more detail.

-       Cabanek et al., 2020. Biophilic streets: a design framework for creating multiple urban benefits.

-      Ko et al. , 2022. A window view quality assessment framework.

-       Berke et al., 2014. Built environment change: A framework to support health-enhancing behavior through environmental policy and health research.

#4: S3: I would like to recommend including more figure images to this section. The information outlined is certainly interesting, but there are no breaks between any paragraphs until page 10 of 20. This doesn’t give readers any chance to process information between the different sections. In particularly, I felt that section 3 provided an opportunity to showcase examples of urban regeneration, and space reproduction using before, and after, photographs for some examples highlighted by the author (e.g., “Chu Milky Street” mentioned on P8, L387-388.) Visualizing the effects that the theoretical framework may have may help readers see what potential practical implications this work can have.

#5: S3.1: This section could also be better structured to refrain from large amounts of information being presented without any text breaks. Currently, information spans from P10 to 16 without any figures, tables, or sub-headings, used to structure the information. Since this part of the paper refers to the development of the framework, which is the core aspect and should be given high priority, please consider changing this to a main section heading (e.g., section 4 (not 3.1)), and more importantly, create subheadings to help organize the text into its most essential elements. The discussion section (P16, L784-786) specifies that the framework can be divided into four core elements, which could be used as the subheading titles. It might be helpful also summarize the four core elements in a table, similar to Table 4 in Cabanek et al. (2020), who had applied this approach to their framework. However, the author may propose an alternative structure to better suit the information already presented in this section.

#6: As the paper stands, there are no conclusions drawn from this work. This could be included at the finale of the paper, after the discussion section, elucidating the main underlying points the author would like to convey to readers. I think this should be connected back to the main aim of the study more directly, since I wasn’t able to find how the theoretical framework clearly responded to this part.

Round 2

Reviewer 2 Report

Thank you for addressing my earlier comments. I felt that there was an improvement to this work, and this is evident within the amendments and responses provided to the feedback given. In particularly, I appreciated the new figures that were added to this paper. I have suggested further comments to help provide additional improvements to this work. My first comment, to the methodology, is rather minor, but I thought would be useful to address. Most importantly, I think that the discussion and conclusion sections are too similar, and there is not a lot of distinction between the two parts. Detailed comments can be found below, which I hope the author finds useful:

#1: Thank you for addressing my earlier point. I felt there was still some room for improvement. Please include references to support the methodology used to frame and identify literature, relevant to the scope of this work. These can be added to page 3, lines 105-108. The selection of the review types is also similar to other conceptual frameworks, particularly in its development (e.g., extracted from interdisciplinary concepts around key components) and execution (e.g., promoting the practical utility for urban regeneration.) I felt this was important to mention, since many past endeavors are based on theoretical ideas, and are conceptualized into practical frameworks. There are many, and many vastly different, approaches that can be used to response to this requirement, and highlighting other frameworks would be useful to show how viable this approach is over different review styles (e.g., a purely systematic review style.)

#2: Please consider moving the conclusion section to after the discussion, since this communicates across the concluding remarks from the review. The conclusion should also be more succinct, overviewing the most important aspects of this work. The second and third paragraphs, in section 7, read more as discussion points, which could be moved into that section. This also includes areas of future research. Potential limitations could also included here (e.g. new gaps that emerged or still could not be answered.) The first, second, and third points, found in the opening paragraph of section 7, could be used as main takeaway messages to help frame the conclusions. The author could consider including these as separate paragraph points to help highlight these to the readers, and expanding upon these to convey across what was learned from this review.
